# Autophagy Activated by Peroxiredoxin of *Entamoeba histolytica*

**DOI:** 10.3390/cells9112462

**Published:** 2020-11-12

**Authors:** Xia Li, Yuhan Zhang, Yanqing Zhao, Ke Qiao, Meng Feng, Hang Zhou, Hiroshi Tachibana, Xunjia Cheng

**Affiliations:** 1Department of Medical Microbiology and Parasitology, School of Basic Medical Sciences, Fudan University, Shanghai 200000, China; 16111010006@fudan.edu.cn (X.L.); 17111010073@fudan.edu.cn (Y.Z.); 19111010064@fudan.edu.cn (Y.Z.); mengfeng@fudan.edu.cn (M.F.); 18111010069@fudan.edu.cn (H.Z.); 2Institute of Metabolic & Integrative Biology, Fudan University, Shanghai 200438, China; qiaoke@fudan.edu.cn; 3Department of Infectious Diseases, Tokai University School of Medicine, Isehara, Kanagawa 259-1193, Japan; htachiba@is.icc.u-tokai.ac.jp

**Keywords:** *Entamoeba histolytica*, autophagy, peroxiredoxin, macrophage, innate immunity, toll-like receptor 4

## Abstract

Autophagy, an evolutionarily conserved mechanism to remove redundant or dangerous cellular components, plays an important role in innate immunity and defense against pathogens, which, in turn, can regulate autophagy to establish infection within a host. However, for *Entamoeba histolytica*, an intestinal protozoan parasite causing human amoebic colitis, the interaction with the host cell autophagy mechanism has not been investigated. In this study, we found that *E. histolytica* peroxiredoxin (Prx), an antioxidant enzyme critical for parasite survival during the invasion of host tissues, could activate autophagy in macrophages. The formation of autophagosomes in macrophages treated with recombinant Prx of *E. histolytica* for 24 h was revealed by immunofluorescence and immunoblotting in RAW264.7 cells and in mice. Prx was cytotoxic for RAW264.7 macrophages after 48-h treatment, which was partly attributed to autophagy-dependent cell death. RNA interference experiments revealed that Prx induced autophagy mostly through the toll-like receptor 4 (TLR4)–TIR domain-containing adaptor-inducing interferon (TRIF) pathway. The C-terminal part of Prx comprising 100 amino acids was the key functional domain to activate autophagy. These results indicated that Prx of *E. histolytica* could induce autophagy and cytotoxic effects in macrophages, revealing a new pathogenic mechanism activated by *E. histolytica* in host cells.

## 1. Introduction

*Entamoeba histolytica* is a protozoan parasite that causes human amoebic colitis and amoebic liver abscess (ALA); it infects 50 million people annually, causing 40,000–100,000 deaths [1]. People are infected by ingesting food and water contaminated by amoeba cysts. Approximately 90% of infected individuals are asymptomatic but in 10%, amoeba trophozoites, driven by unknown stimuli, can penetrate the mucosal barrier of the colon and invade the intestinal lamina propria, leading to amoebic diarrhea and colitis, or even ALA if trophozoites disseminate through the portal circulation [2]. Peroxiredoxins (Prxs) are an evolutionarily conserved family of ubiquitously expressed antioxidant enzymes, which can effectively reduce peroxides, including hydrogen peroxide (H_2_O_2_) and peroxynitrite (ONOO^−^) [3]. As a facultative anaerobic organism, *E. histolytica* requires high amounts of Prx to resist oxidative damage during invasion of host tissues and organs [4,5]. It was shown that ALA was alleviated by Prx downregulation [6], which implicates Prx in the pathogenesis of *E. histolytica* infection. There are more than 20 different transcripts of Prx in *E. histolytica*. In a previous study, we cloned and expressed a Prx of *E. histolytica* (XP_648522.1), which represents a typical 2-Cys Prx containing two catalytic cysteine residues in the active sites [7]. Considering that human Prx-1, which is also a typical 2-cys-Prx, has been shown to be secreted and to bind toll-like receptor 4 (TLR4) on macrophages to promote the inflammatory response [8], it is important to find out whether the Prx of *E. histolytica* can act as a pathogen-associated molecular pattern (PAMP) motif.

Autophagy is a housekeeping process necessary to remove damaged or redundant cellular components; it is executed through the function of autophagosomes and lysosomes and is essential for the maintenance of the metabolic balance in eukaryotic cells during nutritional restriction, infection, or other physiological/pathological conditions [9]. There are three forms of autophagy: microautophagy, chaperone-mediated autophagy, and macroautophagy [10]. Macroautophagy (hereinafter referred to as autophagy) is an evolutionarily conserved process involved in the stress response, during which it eliminates redundant or potentially dangerous cytosolic entities such as damaged mitochondria or invading pathogens [10,11]. Furthermore, it has been found that autophagy plays an important role in innate immunity, a process activated by TLRs and other pattern recognition receptors to eliminate pathogens [12].

Several protozoan parasites have been shown to interact with the autophagy machinery in host cells [9,13]. Thus, it has been reported that the number of light chain 3 (LC3)-positive phagophores in host cells gradually increases after infection with *Trypanosoma cruzi*, indicating a specific molecular exchange between parasites and host cells [14]. In the process of *Leishmania donovani* infection, macrophage autophagy was inhibited at the early stage but was activated 24 h after invasion, which proved to be beneficial for parasite survival [15]. However, the relationship between *E. histolytica* and macrophage autophagy has rarely been reported [16].

The aim of this study was to investigate whether *E. histolytica* Prx (*Eh*-Prx) is involved in the regulation of autophagy in macrophages. The results indicate that the recombinant protein (*Eh*-rPrx) could induce macrophage autophagy accompanied by autophagy-dependent cell death (ADCD) and that the C-terminal part of *Eh*-rPrx containing 100 residues is the principal domain responsible for autophagy activation. This study reveals, for the first time, that *Eh*-Prx is capable of inducing autophagy in immune cells, which is important for further understanding of *E. histolytica* pathogenesis and the role of autophagy in infection.

## 2. Materials and Methods

### 2.1. Ethics Statement

All animal experiments in our study were conducted in strict accordance with the Regulations for the Administration of Affairs Concerning Experimental Animals (1988.11.1) and were approved by the Institutional Animal Care and Use Committee (IACUC) (Permit Numbers: 20160225-097). All efforts were made in our study to minimize the suffering of animals.

### 2.2. Cell Culture

RAW264.7 cells (ATCC, BFN607200597) were cultured in Dulbecco’s modified Eagle’s medium (DMEM) (Corning, Manassas, USA, #10-013-CV) containing 10% (*v*/*v*) fetal bovine serum (HyClone, Beijing, China, SH30396.03), penicillin–streptomycin (100 U/mL; Gibco, #15140122) at 37 °C in a 5% CO_2_ incubator. Trophozoites of *E. histolytica* HM-1: IMSS were cultured in YIMDHA-S medium with 10% (*v*/*v*) heat-inactivated adult bovine serum (Bersee, Beijing, China, B1059) at 36.5 °C.

### 2.3. Expression of Recombinant Proteins

Trophozoites of *E. histolytica* in the logarithmic growth phase were collected. Total RNA was obtained from trophozoites by the RNeasy Maxi Kit (QIAGEN, Hilden, Germany, #75162). Complementary DNA was then acquired from the total RNA above by the PrimeScript™ 1st Strand cDNA Synthesis Kit (TaKaRa, Kusatsu, Japan, # 6110A) and used as a template to generate Prx fragments by PCR. Whole-length Prx (XP_648522.1) was amplified using the following primers:

sense-CCCATATGTCTTGCAATCAACAAAAAGAGT and

antisense-CCGGATCCTTTTAATGTGCTGTTAAATATT.

Three fragments, each encoding a 100-residue part of Prx, were obtained using the following primer pairs:

N-terminal sense-CCCATATGTCTTGCAATCAACAAAAAGAGT and

antisense-CCGGATCCTTTTATTGTCCTGCAAGTTCACTAT;

middle sense-CCCATATGAAGTTGACATTCCCATTAGTATCA and

antisense-CCGGATCCTTTTAATGTGCTGTTAAATATT;

C-terminal sense-CCCATATGGGAAAATATGTTGTATTGTTGTTT and

antisense-CCGGATCCTTTTAATCATCAATGATGACATATC.

The protocol for plasmid construction and protein expression was published previously [7]. His-Bind Resin (Novagen, San Diego, CA, USA, #69670) was used to purify the recombinant proteins, which were dialyzed against Tris-HCl (20 mM, pH 8.0), followed by filtration via a 0.22-µm membrane, and tested for endotoxin levels using the Toxin Sensor^TM^ Gel Clot Endotoxin Assay Kit (Genscript, Nanjing, China, L00351). The endotoxin levels of the recombinant proteins correspond to the national standard of the People’s Republic of China for medical products (GB/t14233.2-2005).

### 2.4. Detection of Autophagosomes by Laser Confocal Microscope

RAW264.7 cells were harvested in the logarithmic growth phase and 10^6^ cells were seeded on sterilized cover slides (22 × 22 mm) placed in 35-mm cell culture dishes (Corning). After cell adhesion, the supernatant was discarded, cells were washed twice with warm phosphate buffered saline (PBS), incubated with *Eh*-rPrx (5 μg/mL) for 24 h, fixed with paraformaldehyde (4% in PBS), and permeabilized with Triton X-100 (0.2%). After blocking with bovine serum albumin (3% in PBS), cells were treated with anti-LC3A/B monoclonal antibody (1:200; Cell Signaling Technology, Cambridge, MA, USA, #12741) and with Alexa Fluor 488-labelled goat anti-rabbit IgG (H + L) (1:200; Invitrogen, A11008). After counterstaining with 0.5 μg/mL 4′,6-diamidino-2-phenylindole (DAPI), cells were preserved at 4 °C in a dark place temporarily or immediately observed under a laser confocal microscope (SP8, Leica, Wetzlar, Germany).

### 2.5. Construction of a Mouse Model of Peritonitis

Female C57 BL/6 mice (10–12 weeks old) were purchased from Shanghai Slake Laboratory Animal Company. Peritonitis was induced by intraperitoneal injection of *Eh*-rPrx (100 µg) in 500 μL Tris-HCl (pH 8.0); the control group received the same volume of Tris-HCl (pH 8.0). After 12 h of treatment, mice were injected with excess of 1% sodium pentobarbital and blood was collected from the heart. Mice were then intraperitoneally injected with 5 mL of cold PBS and massaged gently for 5 min and peritoneal cells were then collected as peritoneal macrophages.

### 2.6. Detection of Autophagy by Western Blotting

RAW264.7 cells were harvested in the logarithmic growth phase and seeded in 6-well plates (10^6^ cells/well). After adhesion, cells were treated with serum-free medium containing recombinant protein (5 µg/mL) or Tris-HCl (pH 8.0) for 24 h, collected, lysed, and analyzed for protein expression by Western blotting using the following primary antibodies: anti-β-actin (ab8227, Abcam, Cambridge, UK), anti-Beclin-1, and anti-LC3A/B (Cell Signaling Technology, #3495 and #12741, respectively). After incubation with secondary horseradish peroxidase (HRP)-conjugated goat anti-rabbit IgG (H + L) (Abcam, ab6721), ImageJ 1.52a (Wayne Rasband National Institutes of Health, Bethesda, MD, USA) was used for signal intensity evaluation.

### 2.7. Analysis of Cell Morphology by Differential Interference Contrast Microscopy

RAW264.7 cells were collected in the logarithmic growth phase and placed in 4-chamber glass bottom 35-mm dishes (In Vitro Scientific, Sunnyvale, CA, USA, D35C4-20-1-N) with 2.5 × 10^5^ cells/chamber. After the cells adhered and fully stretched, serum-free medium containing *Eh*-rPrx was added and real-time changes in cell morphology were immediately observed under a differential interference microscope (Carl Zeiss Meditec, Jena, Germany).

### 2.8. Evaluation of Cytotoxicity

RAW264.7 cells in the logarithmic growth phase were placed in 96-well plates (7.5 × 10^4^ cells/well); pre-treated for 2 h with wortmannin (10 nM, 100 nM, 1 μM, and 10 μM; APExBIO, A8544); treated with *Eh*-rPrx (5 μg/mL) for 12, 24, or 36 h; and analyzed for cell viability using Cell Counting kit-8 (CCK-8; DOJINDO, CK17). Briefly, culture supernatant (100 μL) from each well was mixed with the working solution (100 μL) and incubated in a new 96-well plate protected from light at room temperature for 30 min. After addition of a stop solution (50 μL/well), the absorbance was measured at 450 nm.

### 2.9. Quantification of Autophagy by High-Content Screening Analysis

RAW264.7 cells were harvested in the logarithmic growth phase and placed in 96-well plates (10^5^ cells/well). RAW264.7 cells were pre-treated or not with TLR4 inhibitor TAK242 (1 μM; APExBIO, A3850), incubated with recombinant proteins (5 μg/mL) for 24 h, and stained with monodansylcadaverine (MDC) (100 μM; Sigma-Aldrich, #30432) at 37 °C in the dark for 10 min. The cells were washed gently with warm Ca/Mg-containing Hank’s Balanced Salt Solution (Gibco, #14025-092) and counterstained with Hoechst 33342 (1 μM) at 37 °C in the dark for 5 min. After repeated washing with the same buffer, cells kept in warm buffer were immediately observed and quantified using the Operetta High-content Screening system (Perkin-Elmer, Hopkinton, MA, USA).

### 2.10. SiRNA Treatment of RAW264.7 Cells

TLR4 siRNA (target sequence: CAATTCTGTTGCTTGTATA), TRIF siRNA (target sequence: GGGAAGACCACACCTATAA), MyD88 siRNA (target sequence: GACTGATTCCTATTAAATA), and control siRNA (sequence unknown) were synthesized by Guangzhou RiboBio Co., LTD. RAW264.7 cells in the logarithmic growth phase were placed in a 6-well plate (2.5 × 10^5^ cells/well), treated with siRNA (50 nM) for 24 h, and then incubated with or without recombinant proteins (5 μg/mL) for 24 h.

### 2.11. Enzyme-Linked Immunosorbent Assay

An EIA/RIA 96-well plate (Corning, #3590) was coated with recombinant proteins (0.5 μg/well) in a coating buffer (35 mM NaHCO_3_, 15 mM NaCO_3_) overnight at 4 °C, then washed six times with PBS-Tween 20 (0.05%), blocked with 1% skim milk in PBS for 1 h, incubated with diluted (1:50, 1:100, 1:200) mouse ascites containing anti-*Eh*-Prx mAb 4G6 [17] for 1 h, washed as above, and incubated with HRP-conjugated goat anti-mouse IgG heavey chain and light chain (H + L) for 1 h. After washing as described, chromogenic solution (24 mM C_6_H_5_Na_3_O_7_, 50 mM Na_2_HPO_4_, 0.05% H_2_O_2_, and 4 mM o-phenylenediamine) was added to the wells for 30 min. After terminating with 1 M H_2_SO_4_, the absorbance of the wells was detected at 490 nm.

### 2.12. Statistical Analysis

The data are presented as the mean ± standard error of the mean (SEM). GraphPad Prism 5 (GraphPad Software, Version 5.01, USA) was used for statistical analysis. The difference between treatment groups was analyzed by Student’s *t*-test and a *p*-value < 0.05 was considered statistically significant.

## 3. Results

### 3.1. Autophagosome Formation Was Induced by Eh-rPrx

Maturation of LC3 to the conjugated LC3-II form at the expansion stage of the phagophore is one of the markers of autophagosome formation [10]. After 24-h treatment with *Eh*-rPrx, aggregation of large LC3 particles in RAW264.7 cells could be observed by laser confocal microscopy, indicating the formation of autophagosomes (Figure 1A). In order to investigate the function of *Eh*-rPrx in vivo, a mouse peritonitis model was established by intraperitoneal injection of *Eh*-rPrx. Western blotting analysis of peritoneal macrophages after 12 h revealed that the ratio of LC3-II to LC3-I in the *Eh*-rPrx group was significantly increased compared with the control group (*p* < 0.05, Figure 1B), thus confirming autophagosome formation. These results suggested that *Eh*-rPrx could induce autophagy in macrophages.

### 3.2. Eh-rPrx Caused Autophagy-Dependent Cell Death

Real-time differential interference microscopy analysis showed that *Eh*-rPrx induced changes in RAW264.7 cell morphology: cells became larger, the number of intracellular vacuoles increased, and cell borders gradually became less distinct (Figure 2A). Furthermore, cell viability decreased over the time of *Eh*-rPrx treatment (Figure 2B). Thus, over 80% cells died 48 h after exposure to *Eh*-rPrx; however, an autophagy inhibitor wortmannin (10 μM) reduced this effect. These data indicated that *Eh*-rPrx caused macrophage cell death partly through induction of autophagy.

### 3.3. Eh-rPrx Activated Autophagy through the TLR4–TRIF Pathway

Quantitative assessment of *Eh*-rPrx-induced mean fluorescence intensity (MFI) of MDC by high-content screening analysis did not reveal any difference between the treatment and control groups; however, intracellular aggregation of MDC-labelled vesicles was observed in the treatment group (Figure 3A). Measurement of the fluorescent spot area revealed that it was larger in the treatment group than in the control group, suggesting the formation of autophagosomes, which could be significantly antagonized by a TLR4 inhibitor (*p* < 0.001) (Figure 3A). To further verify the molecular mechanisms of autophagy induction by *Eh*-rPrx, we performed RNA interference experiments using TLR4-, MyD88-, and TRIF-specific siRNAs. The results showed that autophagy was activated by *Eh*-rPrx, mostly through the TLR4–TRIF pathway; the TLR4–MyD88 pathway implicated in NLR family pyrin domain containing 3 (NLRP3) inflammasome activation [18] could also play a role but the effect was not statistically significant (Figure 3B).

### 3.4. Eh-rPrx Activated Autophagy through Its C-Terminal Domain

Our previous results showed that the N-terminal of *Eh*-rPrx (XP_648522.1) was cysteine-rich and was longer than that of non-pathogenic *Entamoeba moshkovskii* Prx [7]. To determine whether the N-terminal of *Eh*-rPrx was required for autophagy activation, we cloned, expressed, and purified three recombinant fragments of *Eh*-rPrx, each comprising 100 residues of the N-terminal, middle, and C-terminal domains, respectively. Among them, the *Eh*-rPrx C-terminal fragment demonstrated a stronger effect on autophagy induction (Figure 4A,B). In addition, *Eh*-rPrx-activated autophagy was reduced by mAb 4G6 (Figure 4C), whose recognition site was confirmed by ELISA to be located at the *Eh*-rPrx C-terminal (Figure 4D). These results suggested that the 100-residue C-terminal domain of *Eh*-rPrx was crucial for autophagy activation.

## 4. Discussion

Host innate immunity acts against *E. histolytica* infection as a double-edged sword. Inducible nitric oxide synthase (iNOS) plays an important role in macrophage-mediated killing of *E. histolytica* and mice lacking iNOS are prone to ALA and hepatocyte apoptosis [2]. However, the formation of ALA is not directly caused by amoebae but is rather related to the innate immune response of the host [19]. Therefore, it is important to study the interaction between *E. histolytica* and macrophages.

Prx can be abundantly produced following parasitic invasion of tissues and organs [20]. The interaction between Prxs of various parasites and macrophages has been reported in several studies. Thus, the Prx of *Plasmodium berghei* could act as a PAMP binding to TLR4 on the macrophage surface, which promoted inflammation [21], whereas the Prx of *Schistosoma mansoni* and *Fasciola hepatica* activated Th2 cells and macrophages [22]. In this study, we showed that the Prx of *E. histolytica* could also act as a PAMP that activated macrophage autophagy through the TLR4–TRIF pathway.

Microtubule-associated protein 1A/1B-LC3 is a key biomarker of autophagy in mammalian cells. During autophagy, ATG4, ATG7, and ATG3 cooperate to cleave the precursor of the LC3-like protein into a mature form [10]. Cytosolic LC3-I is conjugated with phosphatidylethanolamine to form LC3-II, which is recruited to and integrated into the autophagosome membrane; autophagosomes then fuse with lysosomes to form autolysosomes [23]. There is no “gold standard” for measuring autophagy [24,25], which is traditionally evaluated based on LC3 expression determined by immunoblotting and fluorescence microscopy [26]. Immunoblotting is used to detect the transformation of LC3 from Type I to Type II, which is related to the number of autophagosomes [27], whereas immunofluorescence can identify cells undergoing autophagy by labeling LC3 spots [23]. In this study, we observed that treatment with *Eh*-rPrx stimulated the aggregation of LC3 spots in RAW264.7 cells, indicating the formation of autophagosomes (Figure 1A), and significantly increased the LC3-II/LC3-I ratio in mouse peritoneal macrophages, suggesting the upregulation of autophagosome production in vivo (Figure 1B). The complex containing Beclin-1 and VPS34, VPS15, AMBRA1, and/or UVRAG has Class III phosphoinositide 3-kinase activity, which can produce phosphatidylinositol (3,4,5)-trisphosphate to trigger autophagosome nucleation [10]. Therefore, some studies have shown that activation of autophagy is accompanied by the upregulation of Beclin-1 [28,29]. Consistent with these results, we also observed an increase in Beclin-1 expression in RAW264.7 cells after treatment with *Eh*-rPrx (Figure 3B).

Cell death can be classified into three types based on the molecular mechanism and cell morphology: Type 1 or apoptosis, Type 2 or autophagy, and Type 3 or necrosis [30]. Among these, autophagy is characterized by extensive intracellular vacuolization [30], which is consistent with our results (Figure 2A). Some studies have found that the loss of Bcl-2 (a Beclin-1 inhibitor) can lead to excessive autophagy, resulting in cell death [31]. Although cell death is usually accompanied by autophagy, ADCD can be considered only if it is suppressed by autophagy inhibitors (such as 3-methyladenine and wortmannin) or genetic ablation of essential autophagy genes [32]. In our study, the reduced viability of *Eh*-rPrx-treated RAW264.7 macrophages could be partly attributed to ADCD, as it was sensitive to wortmannin (Figure 2B).

It has been shown that autophagy is the downstream effect of TLR signaling [12]. In our study, the TLR4 inhibitor TAK242 significantly reduced the area of autophagy spots, indicating that *Eh*-rPrx is a TLR4 ligand (Figure 3A), which is consistent with similar results for *P. berghei* Prx [21] and human Prx-1 [8]. Autophagy induced by *Eh*-rPrx was shown to occur mainly through the TLR4–TRIF pathway (Figure 3B). The TLR4–MyD88 pathway, which is involved in activation of the NLRP3 inflammasome related to autophagy [18], may also contribute to *Eh*-rPrx-dependent stimulation of autophagy in macrophages, although the results did not reach statistical significance. The 100-residue C-terminal domain of *Eh*-rPrx was the key site responsible for *Eh*-rPrx binding to macrophage TLR4 and appeared to be also the recognition site of the Prx-specific monoclonal antibody (Figure 4).

In conclusion, this is the first study to show that Prx of *E. histolytica* activates autophagy in macrophages through the TLR4–TRIF pathway. The pathogenicity of *E. histolytica* is known to depend on its contact with macrophages; however, the results of this study show that *E. histolytica* is capable of inducing autophagy and cell death in a parasite–macrophage contact-independent manner through secretion of Prx. Our study provides further insights into the molecular mechanism underlying *E. histolytica* pathogenicity, which could aid in the identification of potential drug targets in the amoeba.

## Figures and Tables

**Figure 1 cells-09-02462-f001:**
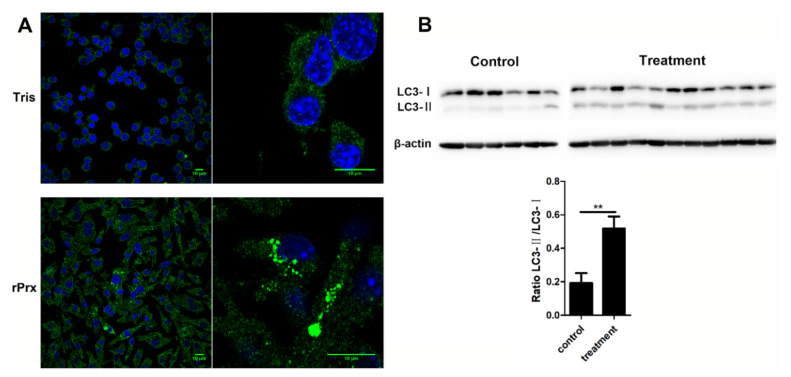
Autophagosome formation was induced by recombinant *E. histolytica* peroxiredoxin (*Eh*-rPrx). (**A**) *Eh*-rPrx promoted light chain 3 (LC3) aggregation in macrophages. RAW264.7 cells were incubated with *Eh*-rPrx (5 μg/mL) for 24 h and analyzed by immunofluorescence and confocal microscopy. Scale bar: 10 μm. (**B**) *Eh*-rPrx could induce autophagy in vivo. Mice were intraperitoneally injected with *Eh*-rPrx (100 μg) and peritoneal macrophages were analyzed after 12 h for LC3 expression by Western blotting. A representative image is shown. The data are expressed as the mean ± standard error of the mean (SEM) (*n* = 6 or 11); ** *p* < 0.01 by Student’s *t*-test.

**Figure 2 cells-09-02462-f002:**
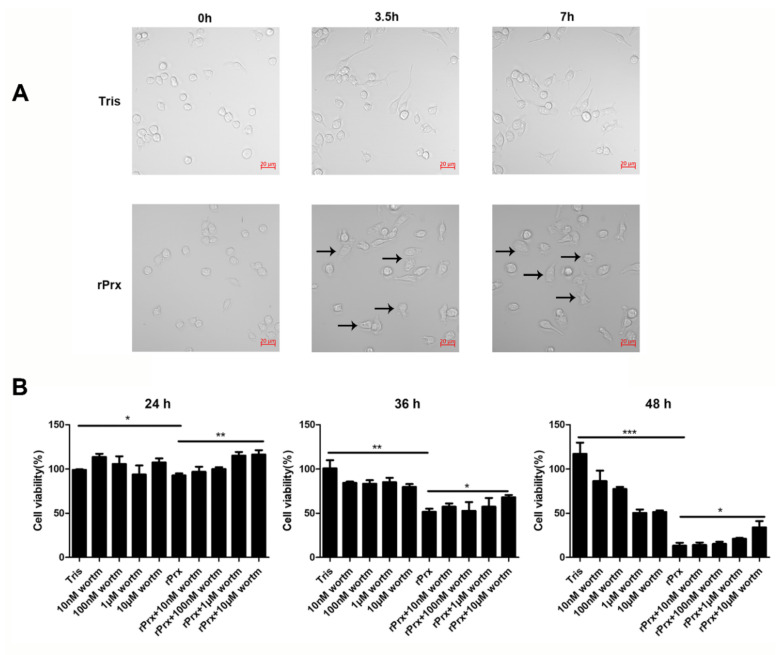
*Eh*-rPrx caused autophagy-dependent cell death. (**A**) *Eh*-rPrx treatment induced morphological changes in RAW264.7 cells. Cells were treated with serum-free Dulbecco’s modified Eagle’s medium (DMEM) containing *Eh*-rPrx (5 μg/mL) and analyzed by differential interference microscopy at the indicated times. Cells treated with *Eh*-rPrx showed morphology alterations: large area, vacuolation, and indistinct membranes (black arrows). Scale bar: 20 μm. (**B**) *Eh*-rPrx-induced cell death was autophagy-dependent. RAW264.7 macrophages were pre-treated with wortmannin (10 nM, 100 nM, 1 μM, and 10 μM), then treated with *Eh*-rPrx (5 μg/mL); cell viability was evaluated at the indicated times. The data are expressed as the mean ± SEM (*n* = 5); * *p* < 0.05, ** *p* < 0.01, and *** *p* < 0.001.

**Figure 3 cells-09-02462-f003:**
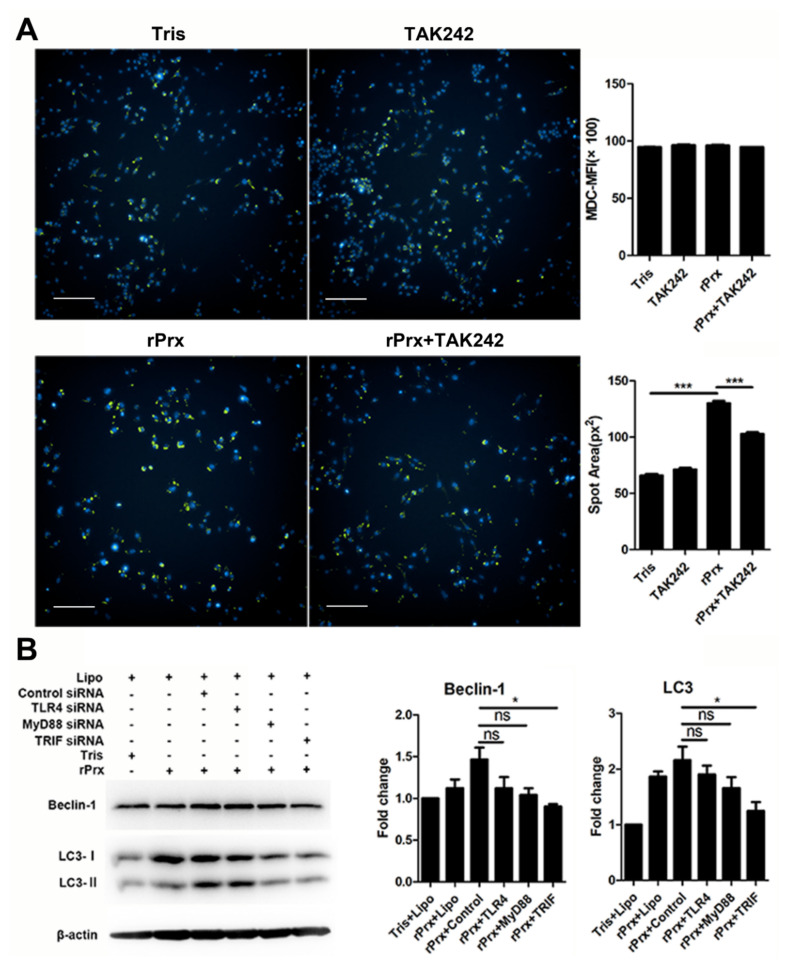
*Eh*-rPrx activated autophagy via the toll-like receptor 4 (TLR4)–TRIF signaling pathway. (**A**) *Eh*-rPrx induced autophagosome formation through TLR4. RAW264.7 cells were pre-treated with the TLR4 inhibitor TAK242, incubated with *Eh*-rPrx for 24 h, and stained with monodansylcadaverine (MDC) (green) and Hoechst 33342 (blue). Tris-HCl (pH 8.0) was used for the negative control. The spot area of MDC fluorescence was quantitatively assessed by high-content screening analysis (*n* = 5). Scale bar: 100 μm. (**B**) Autophagy induction by *Eh*-rPrx occurred through the TLR4–TRIF signaling pathway. RAW264.7 cells were pre-treated with TLR4-, MyD88-, or TRIF-specific siRNAs for 24 h, treated with *Eh*-rPrx (5 μg/mL) for 24 h, and analyzed for the expression of Beclin-1 and LC3 by Western blotting (*n* = 3). Tris-HCl (pH 8.0) was used for the negative control. The data are expressed as the mean ± SEM; * *p* < 0.05 and *** *p* < 0.001 by Student’s *t*-test; ns, not significant.

**Figure 4 cells-09-02462-f004:**
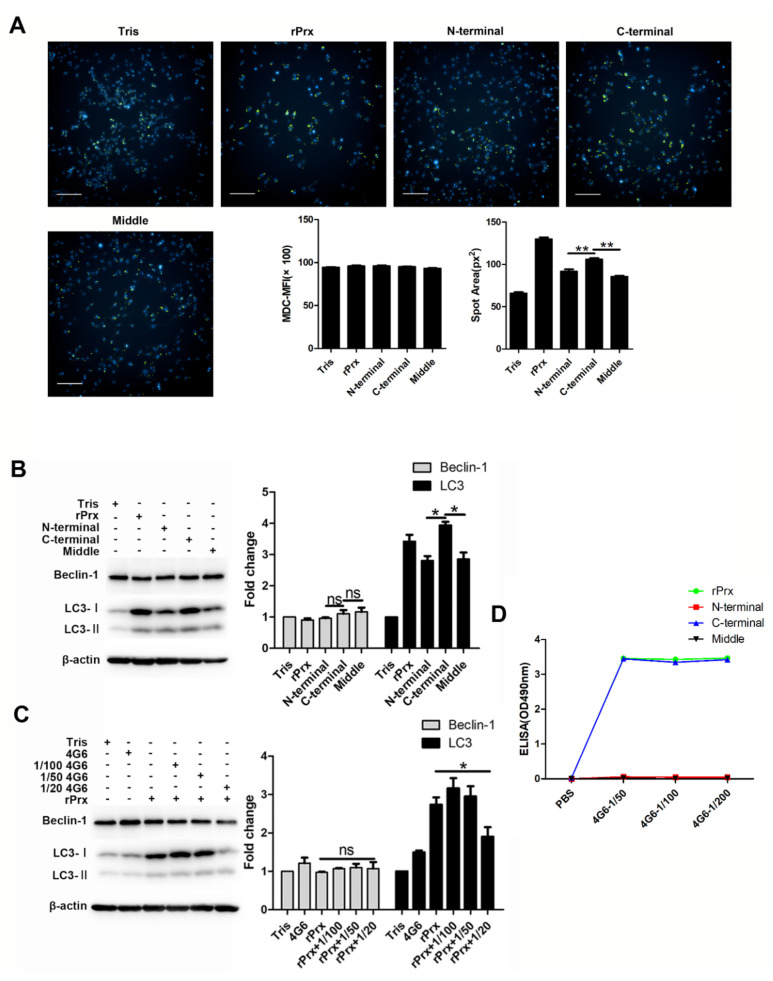
The *Eh*-rPrx C-terminal was the principal domain responsible for autophagy activation. (**A**) The C-terminal of *Eh*-rPrx induced autophagosome formation. RAW264.7 cells were incubated with 5 μg/mL of full-length *Eh*-rPrx or its N-terminal, middle, or C-terminal fragments for 24 h and then stained with MDC (green) and Hoechst 33342 (blue). The spot area of MDC fluorescence was quantified using high-content screening analysis (*n* = 5). Scale bar: 100 μm. (**B**) *Eh*-rPrx C-terminal fragment activated autophagy. RAW264.7 cells were treated as in (**A**) and analyzed for the expression of Beclin-1 and LC3 by Western blotting (*n* = 3). (**C**) Autophagy activation by *Eh*-rPrx was inhibited by mAb 4G6. RAW264.7 cells were pre-treated with different concentrations of mAb 4G6, incubated with *Eh*-rPrx (5 μg/mL) for 24 h, and analyzed for the protein expression of Beclin-1 and LC3 by Western blotting (*n* = 3). (**D**) *Eh*-rPrx C-terminal fragment contained the recognition site for mAb 4G6. The reaction of full-length *Eh*-rPrx or its N-terminal, middle, or C-terminal fragments with mAb 4G6 was analyzed by ELISA. The data are expressed as the mean ± SEM; * *p* < 0.05 and ** *p* < 0.01 by Student’s *t*-test; ns, not significant.

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
