# Peer review of "Autophagy Activated by Peroxiredoxin of Entamoeba histolytica"

_cells, 2020, doi:10.3390/cells9112462_

Round 1

Reviewer 1 Report

REPORT of article: Xia Li et al., 2020.

Xia Li and colleagues report that the recombinant protein peroxiredoxin from E. histolytica induces autophagy in macrophages. They demonstrate in vivo and in vitro the capability of Eh-rPrx for induce phagosomes formation in vivo (peritoneal mouse macrophages), and in vitro in RAW264.7 cells, also perform experiments to demonstrate that Eh-rPrx induce phagosomes formation through TLR4 signaling.

However, there are some facts that are not clear, for example, in figure 2 the effect of Eh-rPrx on RAW264.7 cells, morphology is not described and not identifying on the figures, so, is needed arrows or something else for signal the morphology alteration on cells. Figures have a bad quality.

In figure 3, what does it mind TRIS?. it is no defined in legend to figure 3.

In general, there are few grammatical mistakes as when refer “Western blotting nalysis”, the correct word is “analysis”.

The manuscript can be considerate to be published in cells, if they are capable for arrange the mistakes described before.

Author Response

Dear Reviewer 1, 

    Thanks very much for your comments. A point-by-point response has been constructed, please see the attachment.  

Reviewer 2 Report

In this study, authors investigated the interaction between E. histolytica and macrophages in order to understand the process of autophagy in host cells. They have been successful in identifying a novel pathogenic mechanism of autophagy in host cells activated by E. histolytica. They showed that the E. histolytica antioxidant enzyme peroxiredoxin (Prx) plays a critical for ameba survival during invasion of host tissues. It could activate autophagy in macrophages. With the help of immunofluorescence staining experiments, authors showed that the macrophages treated with recombinant E. histolytica Prx for 24 h could induce the formation of autophagosomes in mice. Authors were able to decipher the mechanism of autophagy using the RNA interference experiments. This revealed that the Prx-induced autophagy occurs mostly through the TLR4-TRIF pathway. They also showed that the C-terminal 100 amino acid residues of Prx was key functional domain to activate autophagy. This reviewer thinks this is a significant achievement in understanding the role of autophagy in E. histolytica pathogenesis.

One of my concerns is that some of the experiments could use the most homologous recombinant Prx enzyme from E. dispar (preferred; or from E. moshkovskii). This would provide further evidence that only E. histolytica Prx, not E. dispar Prx, is capable of inducing autophagy in hosts. Another concern is the use of English language and grammar in the manuscript. For example, there has been excessive use of semicolons in the “Abstract” - semicolon has been used 10 times in 7 full sentences! Some of the sentences are very long.

My other minor comments are:

  1. Page-1/Abstract: In the second line, what does the word “which” refer to?
  2. Page-1/Abstract: In the last sentence, a period is missing.
  3. Page-2/Introduction: In the penultimate paragraph it says, “However, the relationship between E. histolytica and macrophage autophagy has not been reported.” While this is correct, but it is also true that others also suspected autophagy in hosts by E. histolytica. Betanzos et al, 2013 showed that at early time of interaction, E. histolytica increases the incidences of apoptosis and autophagy. I think, this work may be mentioned in the “Introduction”. See the reference: PLoS ONE 8(6): e65100. doi:10.1371/journal.pone.0065100
  4. Page-5/Results/3.1. Autophagosome Formation was Induced by Eh-rPrx: In line 5, change “nalysis” to ‘analysis’.
  5. Page-6/Figure 2: Regarding the Figure 2A, in Page-5 it says, “cell morphology: cells became larger, the number of intracellular vacuoles increased, and cell borders gradually became less distinct (Figure 2A)”. However, it is unclear from the Figures presented here. This is partly because of small size of each individual Figures. Consider showing just 3 time points (such as 0h, 3.5h and 7h) as representatives of events that are taking place instead of 15 time points in control and test groups. Some of the key morphology changes may be included in the Figure legend. Also, in the line-5 of Figure legend, change “5 μg / ml” to ‘5 μg/ml’.
  6. Page-7/Figure 3: Define the size represented by the white bars in Figure 3A.
  7. Page-8/3.4. Eh-rPrx Activated Autophagy through Its C-Terminal Domain: The pathogenicity of Entamoeba moshkovskii remains elusive. One study showed that E. moshkovskii can cause diarrhea in children, and cause diarrhea and colitis in mouse model of amebiasis (see reference: Shimokawa et al, 2012 J Infect Dis. 2012 Sep 1;206(5):744-51. doi: 10.1093/infdis/jis414. Epub 2012 Jun 21. PMID: 22723640). Therefore, a more realistic comparison would have been with that of an E. dispar Prx sequence. As mentioned earlier, an E. dispar Prx could work as a preferred (negative) control for some of the experiments performed in this study.

Author Response

Dear Reviewer 2, 

    Thanks very much for your comments. A point-by-point response has been constructed, please see the attachment.  
